# Cardiac Protection of a Novel Lupane-Type Triterpenoid from Injuries Induced by Hypoxia–Reperfusion

**DOI:** 10.3390/ijms23169473

**Published:** 2022-08-22

**Authors:** Beibei Guo, Jiaxin Cao, Yi Liu, Yuhang Wang, Yi Qian, Guangtong Chen, Weizhong Zhu

**Affiliations:** Department of Pharmacology, School of Pharmacy, Nantong University, Nantong 226001, China

**Keywords:** triterpenoids, cardiac protection, oxidative stress, PI3K/Akt

## Abstract

Myocardial ischemia–reperfusion injury (MIRI) leads to cardiac remodeling and heart failure associated with acute myocardial infarction, which is one of the leading causes of death worldwide. Betulinic acid (BA), a widely distributed lupane-type triterpenoid, has been reported to possess antioxidative activity and inhibit apoptosis in MIRI. Due to the low bioavailability and water insolubility of BA, a previous study found a series of BA-derivative compounds by microbial transformation. In this study, we observe whether there are anti-MIRI effects of BTA07, a BA derivative, on cardiac injuries induced by hypoxia/reoxygenation (H/R) in adult rat cardiomyocytes in vitro and in Langendorff-perfused hearts ex vivo, and further explore its mechanism of cardioprotection to find more efficient BA derivatives. The hemodynamic parameters of isolated hearts were monitored and recorded by a Lab Chart system. The markers of oxidative stress and apoptosis in isolated hearts and adult rat cardiomyocytes (ARCMs) were evaluated. The expression levels of B-cell lymphoma 2 (Bcl-2), Bcl-2-associated X (Bax), protein kinase B (Akt) and phospho-Akt (pAkt, Ser473) induced by H/R were detected via Western blot. The Langendorff experiments showed that BTA07 improves hemodynamic parameters, reduces myocardium damage and infarct size, inhibits levels of myocardial tissue enzymes lactate dehydrogenase (LDH) and creatine kinase (CK) in the coronary outflow and reduces oxidative stress and the activation of caspase-3 in the myocardium. In vitro, BTA07 reduced cell death and caspase-3 activation and inhibited reactive oxygen species (ROS) generation. Furthermore, the protective effects of BTA07 were attenuated by inhibition of the PI3K/Akt signaling pathway with LY294002 in ARCMs. BTA07 protects ARCMs and isolated hearts from hypoxia–reperfusion partly by inhibiting oxidative stress and cardiomyocyte apoptosis.

## 1. Introduction

Natural products, such as triterpenoids [1], have emerged as potential candidates for the treatment of myocardial ischemia–reperfusion injury (MIRI) due to their wide spectrum of biological activity and low toxicity [2]. Pentacyclic triterpenoids (PTs) are an important class of natural products composed of six isoprene units [3]. They are generally present in higher plants and have a broad spectrum of beneficial biological activities, including antioxidative, anti-inflammatory, antiapoptotic and antitumor qualities [4]. PTs mainly contain lupane, oleanane, ursane or friedelane as a primary skeleton [5]. Among them, BA, a lupane-type pentacyclic triterpene, is widely distributed in birch and plane trees and exhibits an array of pharmacological activities [6]. The potential protective effects of BA on ischemia–reperfusion-related diseases have also been confirmed in cerebral ischemia–reperfusion [7]. Recent experimental studies indicate that BA can protect against MIRI in H9c2 cells [8] and in rat hearts [9]. However, its poor solubility and low bioavailability limits its clinical application. In this study, a novel lupane-type triterpenoid known as3-oxo-23-acetoxy-7β-hydroxylup-20(29)-en-28-oic acid (BTA07; Figure 1) was prepared via microbial transformation with a betulinic acid substrate [9].

Ischemic heart disease is one of the leading causes of death worldwide [10]. It has long been recognized that ischemic tissue can survive through the timely restoration of blood flow, known as reperfusion [11]. However, reperfusion causes damage to the myocardium, known as MIRI, through oxidative stress, inflammation and cardiomyocyte apoptosis [12]. Therefore, targeting oxidative stress and cardiomyocyte apoptosis in MIRI is imperative to preventing and treating this disease.

The PI3K/Akt signaling pathway is a signal transduction pathway that participates in the regulation of cell proliferation, oxidative response and cardiac apoptosis [13]. Many studies have recently confirmed that this signaling pathway not only inhibits cardiac fibrosis [14] but also plays an active role in reducing cardiac ischemic–reperfusion injury by inhibiting the NF-kB signaling pathway and related inflammatory factors [15]. Furthermore, it has been reported that the cerebral-ischemia–reperfusion-injury-related effects of BA were closely related to the PI3K/Akt pathway [16].

In this study, both the ischemia–reperfusion (I/R)-induced injuries of the perfused rat heart ex vivo and the hypoxia/reoxygenation (H/R) of cultured adult rat cardiomyocytes (ARCMs) in vitro were induced to investigate whether BTA07 has cardioprotective effects. It was hypothesized that BTA07 alleviates MIRI by suppressing oxidative stress and apoptosis through activation of the PI3K/Akt signaling pathway. BTA07 is a compound that can be used as a substrate in microbial transformation technology applications. Therefore, our findings might provide a new direction for the discovery and development of anti-MIRI drugs.

## 2. Results

### 2.1. BTA07 Ameliorated the Suppressed Contractile Function of Isolated Rat Hearts Subjected to I/R

To confirm the cardioprotective effects of BTA07 on MIRI, various hemodynamic parameters of isolated hearts were collected. As shown in Figure 2, the HR, LVEDP and ±dP/dt max of all groups were monitored and recorded at T_0_, T_1_, T_2_, T_3_ and T_4_. Compared with the normoxia group, the HR and +dP/dt max of isolated hearts significantly decreased, whereas the LVEDP and −dP/dt max significantly increased at T_2_, T_3_ and T_4_ in all I/R groups (*p* < 0.01; two-way ANOVA). Hemodynamic parameters could be improved to varying levels after starting reperfusion, depending on the different doses of BTA07. Among them, 10 μM of BTA07 was most effective in improving the function of isolated rat hearts. More importantly, BTA07 with the same concentration can improve cardiac function better than BA, which is a positive control (*p* < 0.01; two-way ANOVA). As expected, the BA group increased HR, ±dP/dt max and decreased LVEDP. Taken together, the data indicate that BTA07 demonstrated stronger effects in improving the hemodynamic parameters of isolated hearts than those in the positive control BA.

Next, myocardial infarct size, a common index for evaluating myocardial injury [17], was measured by TTC staining. As shown in Figure 3, the I/R group showed a large size of myocardial infarction in the isolated heart. The I/R-induced infarct size was significantly decreased after perfusion with BTA07. Among them, pretreatment with10 μM of BTA07 had the most significant effect in reducing infarct size.

To further observe the histological protective effects of the BTA07 treatment, H&E staining was used to observe the pathological changes of isolated hearts. As shown in Figure 3, the results indicate that compared with the normoxia group, the myocardial structure in the vehicle group is abnormal, evidenced by muscle fibers being irregularly arranged and the presence of fibrous hyperplasia. Perfusion with 10 μM ofBTA07 improved the arrangement of cardiomyocytes. Moreover, myocardial enzyme levels in the coronary outflow, LDH and CK-MB were increased after the I/R injury, but decreased after treatment with different doses of BTA07, as shown in Figure 3.

### 2.2. BTA07 Treatment Suppressed Myocardium Oxidativepotential and Apoptosis in Isolated Hearts

Oxidative stress was assessed by detecting levels of MDA, SOD and GSH-Px in perfused hearts. As shown in Figure 4, levels of MDA post I/R were significantly increased, whereas levels of SOD and GSH-Px were significantly decreased, suggesting that an acute oxidative stress response was caused by I/R in the myocardium. Intervention of BTA07 could dose-dependently reduce the level of MDA and promote the production of SOD and GSH-Px.

To investigate the effect of BTA07 on myocardium apoptosis, we measured caspase-3 activity. Figure 4 shows that I/R significantly induced an increase in caspase-3 activity, whereas the induction was suppressed by BTA07 treatment in a dose-dependent manner. More interestingly, significantlydecreasedcaspase-3 activity was observed in the BTA07 group compared with the BA group at the same concentration (see Figure 4D).

### 2.3. BTA07 Decreased the Cell Death Rate and LDH Release in ARCMs Exposed to H/R

To evaluate the cytotoxicity of BTA07, the ARCMs were incubated with different concentrations of BTA07 (0, 0.63, 1.25, 2.5, 5, 10 and 20 μM). Representative PI/Hoechst-stained photomicrographs of ARCMs with BTA07 treatment are shown in Figure 5A. The results of the PI/Hoechst staining show that BTA07 did not promote cell death until the concentration was 20 μM (Figure 5B). Thus, concentrations of 0.63, 1.25, 2.5, 5 and 10 μM were selected for the following study. Subsequently, we explored the protective effects of BTA07 on ARCMs exposed to H/R. As shown in Figure 5D, the death rate of ARCMs was significantly increased after H/R treatment, whereas cell death was attenuated by BTA07 pretreatment (0.63, 1.25, 2.5, 5 and 10 μM). Furthermore, the level of LDH in the supernatant was noticeably increased in the cultured ARCMs exposed to H/R compared with the normoxia group, implying thatBTA07 dose-dependently prevented the increase in LDH release evoked by H/R stress.

### 2.4. BTA07 Recovered the Phosphorylation of Akt Signaling Suppressed by H/Rin ARCMs

The Akt signaling pathway plays an important role in the development of MIRI [18]. To clarify the mechanism underlying the cardioprotective activity of BTA07, Western blotting was used to detect the level of phosphate in Akt. As presented in Figure 6A, the protein expression levels of p-Akt were significantly suppressed by H/R in ARCMs compared with the normoxia group. BTA07 treatment significantly upregulated the expression levels of p-Akt in ARCMs. To further confirm the role of the PI3K/Akt pathway in BTA07-induced ARCMs, levels of LDH release were detected as cardioprotective indicators. LY294002, a specific PI3K inhibitor, attenuated the phosphorylation of PI3K and Akt. Functionally, as shown in Figure 6B, LY294002abolished the cardioprotective effects of BTA07.

### 2.5. Pharmacological Inhibition of PI3K with LY294002 Abolished BTA07-Induced Antioxidativepotential in ARCMs

Indicators of oxidative stress including ROS, MDA, SOD and GSH-Px in ARCMs were measured after BTA07 treatments as shown in Figure 7. Compared with the normoxia group, levels of ROS and MDA were significantly increased, whereas SOD and GSH-Px were significantly decreased. BTA07 treatment remarkably reversed the H/R-induced changes in these oxidative stress markers, indicating that BTA07 possessed anti-oxidative effects. Furthermore, inhibition of PI3K with LY294002 abolished BTA07-inducedantioxidativepotential. Taken together, BTA07 inhibited the H/R-induced oxidative potential in ARCMs via the PI3K/Akt signaling pathway.

### 2.6. BTA07 Inhibited H/R-Induced Apoptosis in ARCMs via the PI3K/Akt Signaling Pathway

To confirm that the cardioprotection mechanisms ofBTA07 occur by inhibiting H/R-induced myocyte apoptosis via a PI3K/Akt-dependent pathway, we measured expressions of apoptosis-related proteins, Bax and Bcl-2. As shown in Figure 8A,B, H/R caused an increase in Bax expression. In contrast, H/R led to a decrease in Bcl-2 expression. However, the H/R-mediated changes in the expression of Bax and Bcl-2 were restored by the BTA07 treatment. Additionally, inhibition of PI3K with LY294002 masked the induction of the BTA07 treatment on levels of Bcl-2 and Bax.

Figure 8C further shows that H/R induced an increase in caspase-3 activity, whereas induction was suppressed by the BTA07 treatment, which was consistent with the ex vivo data. Inhibition of PI3K with a specific blocker, LY294002, abrogated the BTA07-induced decrease incaspase-3 activity. To further confirm if the ROS-mediated apoptosis was suppressed with treatment of BTA07, caspase-3 activity in cultured ARCMs was detected post treatment with 20 μM of H_2_O_2_ for 1 h. As shown in Figure 8D, exposure to H_2_O_2_ resulted in an increase in caspase-3 activity as expected, but BTA07 indeed diminished the caspase-3 activity evoked by H_2_O_2_. The effects were partly abolished by pretreatment of LY294002. These results suggest that BTA07 protects against H/R- and H_2_O_2_-induced cell injuries.

## 3. Discussion

In this study we found the anti-MIRI effect of BTA07, a BA derivative, using microbial catalysis. As far as we know, this is the first study that has explained the cardiac protection of BTA07 from ischemic-induced injuries.

The Langendorff perfused heart is an ex vivo experimental operation that allows us to observe the effects of drugs on isolated mammalian heart activity [8,19]. This study found that BTA07 ameliorated isolated heart function exposed to I/R in a dose-dependent manner, evidenced through the improvement of hemodynamic parameters, reduction in infarct size and levels of myocardial tissue enzyme and inhibition of oxidative stress and apoptosis. In addition, the H&E staining and TTC results further demonstrate that BTA07 improves myocardial histopathological features. Since coronary flow can affect cardiac oxygen supply, it is worth further exploring whether BTA07 affects vascular resistance, which could be one of the reasons for why it protects the heart from injuries induced by H/R.

In recent years, several other forms of cell death have been identified, suggesting that cells can die in many ways. Apoptosis is characterized by many characteristic morphological changes in the cellular structure and many enzyme-dependent biochemical processes [20]. The result is the elimination of cells from the body with minimal damage to surrounding tissues. Eksioglu-Demiralp et al. found that betulinic acid protects against I/R-induced renal damage and inhibits the apoptosis of leukocytes [21]. In our study, we found that BTA07 decreased cell death rate and LDH release in ARCMs exposed to H/R.

The oxidative stress elicited in tissues and cells exposed to I/R (or H/R) has been connected to a variety of different sources of ROS [22]. The production of oxidative stress can damage the membrane lipids of cardiomyocytes, destroy the function and integrity of cardiomyocytes, trigger myocardial cell apoptosis and death and ultimately lead to cardiomyocyte loss and heart disease. Some studies have shown that the protective effect of betulinic acid on I/R- or H/R-induced cardiomyocytes might be related to its upregulation of antioxidative ability [23]. In this study, we observed that BTA07 reduced the level of MDA, significantly increased levels of SOD and GSH-Px and regulated the activation of caspase-3 in the isolated rat heart during the process of I/R injury in a dose-dependent manner.

ARCMs have been widely accepted as a crucial platform for heart physiology and pathology studies and a significant supplement to in vivo and ex vivo experimental studies. Accordingly, the results in vitro show that the production of oxidative stress, including ROS and MDA induced by H/R in ARCMs, is inhibited by BTA07. Meanwhile, BTA07 modulates the expression of Bcl-2 and Bax as well as the activation of caspase-3. Therefore, it can be concluded that the anti-MIRI effect of BTA07 occurs by attenuating oxidative stress and inhibiting cell apoptosis. In the Langendorff model of IR injury, oxygen concentration in the mitochondria of ischemic tissue is negligible due to the lack of blood supply to isolated ischemic hearts and the very low apparent Km of cytochrome oxidase. Meanwhile, glucose supply is constrained during tissue ischemia [24,25]. However, in cell models of IR injury, the oxygen level of most hypoxic cell incubators is 1–0.1% [26]. Besides this, the residual oxygen present in many cellular hypoxic models maintains oxidative phosphorylation of mitochondria. The loss of intracellular lactate into the culture medium during ischemia allows cells to sustain glycolysis. Using a glucose-free lactate-based buffer in ARCMs models is one way to circumvent this. To overcome these limitations, Gruszczyk et al. cultured isolated adult mouse cardiomyocytes under hypoxic conditions while inhibiting lactate efflux. In this study, the cardioprotective effect of BTA07 was observed in both isolated hearts and ARCMs in a glucose-free culture [27].

Due to the proximity of ARCMs to physiological levels rather than cell lines, in this study we chose ARCMs to address the relationship between BTA07-induced cardioprotective effects and the PI3K/Akt signaling pathway. Hypoxia-inducible factor (HIF)-1 is a key downstream protein that is closely related to oxygen concentration in an environment. It is essential in maintaining stable HIF-1 protein levels under normoxic conditions. Under hypoxic conditions, however, HIF-1 levels rise rapidly. Hypoxia-inducible factor-1α is associated with acute erythropoietin-related hypoxic responses, whereas hypoxia-inducible factor-2α is related to chronic hypoxic reactions [28]. PI3K/Akt signaling plays a crucial role in regulating many physiological functions by activating downstream effectors and regulating cell cycle transition, growth and proliferation. This pathway has been implicated in the pathogenesis of several human diseases by regulating cardiomyocyte size and survival, angiogenic processes and inflammatory responses [29]. Activation of the PI3K/Akt pathway is considered an endogenous regulatory mechanism that promotes cell survival in response to harmful external stimuli [30,31]. Numerous studies have shown that the PI3K/Akt signaling pathway plays a critical role in the pathological process of MIRI. Xin et al. showed that the PI3K/Akt/HSP70 signaling axis participates in anti-MIRI effects via visfatin, resulting in reduced inflammatory and apoptotic factors [32]. Shang et al. demonstrated that activation of the PI3K/Akt/mTOR signaling pathway protects the myocardium in sepsis induced by lipopolysaccharide. Jiao et al. revealed that pretreatment of BA attenuated OGD/R (oxygen and glucose deprivation/reperfusion)-induced neuronal injury in rat hippocampal neurons through activation of the PI3K/Akt signaling pathway. Furthermore, administration of the PI3K inhibitor, LY294002, abolished the cardioprotective efficacy and inhibitory effects of BTA07 on oxidative stress and apoptosis in this study. Therefore, BTA07 treatment decreased H/R-induced cell damage, oxidative stress and apoptosis in a PI3K/Akt-dependent manner in cultured cardiac myocytes. Interestingly, betulinic acid induces apoptosis by regulating PI3K/Akt signaling and mitochondrial pathways in human cervical cancer cells [33] and in differentiated PC12 cells via ROS-mediated mitochondrial pathways [34].Our data suggests that treatment with the BA derivative, BTA07, not only protects ARCMs from H/R stimulation, but also recovers the suppressed phosphorylation of Akt evoked by H/R stimulation. The experiment ex vivo is more complicated, taking into consideration the effects of a tested drug on cardiac myocytes, vascular endothelium, vascular smooth muscle, etc. These will all affect drug-evoked cardiac contractile function and protection. In addition to BAT07′s role in cardiac protection, itis reasonable to suspect that Akt signaling in the vascular endothelium could be a target of BAT07, since in this study we found that Akt signaling was suppressed by hypoxia/reoxygenation, and BTA07 modulated Akt signaling to display its cardiac protection in cultured cardiomyocytes. Thus, it is valuable to further study how BTA07 regulatesPI3K activation, consequently activating Akt, and whetherBTA07 affects Akt signaling in myocardium ex vivo and in vivo.

Beyond PIK3/Akt signaling, Li et al. found that betulinic acid inhibits MAKP and activates the NRF signaling pathway [1,35], and Zhou et al. demonstrated that betulinic acid ameliorates the severity of acute pancreatitis via inhibition of the NF-κB signaling pathway in mice [36]. The presence of the unsaturated carbon–carbon bond, as well as the presence of the carbonyl group within the structure of betulinic acid and BTA07, most likely affects its electrophilic nature. This was later demonstrated using the Keap1/Nrf2 system to test the electrophilic/antioxidative potential of betulinic acid in hypoxia/reoxygenation [1]. Therefore, it has not been ruled out that the Nrf2-mediated effect of BTA07 plays an important role in cellular protection against ischemia–reperfusion injuries (IRI), as seen in the experimental setup in this study with the BTA07 pretreatment before the actual IRI.

Based on all the available results, we propose that BTA07 targets the PIK3/Akt signaling pathway. The Akt was perhaps translocated into the mitochondria, where it modulated ROS production, or into the nucleus to regulate the expression of Bax/Bcl-2. Another possibility is that the original mitochondrial ROS regulate the Bax/Bcl-2 expression by mitochondria–nuclear couple signaling. Another possibility is that BTA07 might affect the expression of antioxidative pathways such as SOD and, consequently, a decrease inROS triggers the expression of Bax/Bcl-2, which is positively related to apoptosis and cardiac remodeling [37,38]. Thus, it is worth further studying the cause–effect relationship between ROS-produced signaling and Bax/Bcl-2, although both ROS and Bax/Bcl-2are regulated by the PI3K/Akt signaling pathway in this study.

There were several limitations to this study. Firstly, we only evaluated the effect of BTA07 on myocardial I/R in vitro and ex vivo. In future, an in vivo animal study will be considered. Secondly, only male rats were used. Cardiac protection of BTA07 might have gender-specific factors. Thirdly, this study does not identify the key chemical structure group in BTA07 for its activation.

## 4. Materials and Methods

### 4.1. Animals

Adult Sprague-Dawley rats (male, 250–280 g) were obtained from the Animal Centre of Nantong University. All of the animal experiments were approved by the Board of Nantong University Animal Care and Use. The production license number is: SYXK9 (Su) 2007-0021.

### 4.2. Reagents

The compound, BTA07, was microbial, transformed by betulinic acid and prepared with a purity of 97.9% [9]. Anti-B-cell lymphoma 2 (Bcl-2, 1:1000, Abcam, Cambridge, UK), anti-Bcl-2-associated X (Bax, 1:1000, Abcam, Cambridge, UK), anti-protein kinase B (Akt, 1:1000, Cell Signaling Technology, Danvers, MA, USA), anti-phospho-Akt (p-Akt, Ser473, 1:1000, Cell Signaling Technology, Danvers, MA, USA) and GAPDH (1:2000, BOSTER, Wuhan, China) were also used. Creatine kinase-MB (CK-MB) was detected using commercial kits according to the instructions from the Nanjing Jiancheng Bioengineering Institute. Lactic dehydrogenase (LDH), malondialdehyde (MDA), total superoxide dismutase (SOD) and glutathione peroxidase (GSH-Px) detection kits, LY294002, propidium iodide (PI), Hoechst 33342 and a caspase-3 activity assay kit were purchased from Beyotime of Nantong (Nantong, China). MitoSOX^TM^ Red mitochondrial superoxide indicator(M36008) was purchased from Thermo Scientific. The kit for triphenyltetrazolium chloride (TTC) and hematoxylin and eosin (H&E) staining was purchased from Servicebio of Wuhan (Wuhan, China).

### 4.3. Isolated and Cultured Adult Rat Cardiomyocytes

Adult rat cardiomyocytes (ARCMs) were isolated from adult Sprague-Dawley rats (male, 250–280g), as previously described [39]. ARCMs were isolated by Langendorff perfusion, digested with Type II collagenase and cultured in the serum-free Medium199 with 100 U/mL penicillin–streptomycin solution for 2–4 h before subsequent experimentation. Medium 199 was composed of Medium 199 (Sigma, St. Louis, MO, USA), 5 mM creatine, 5 mM taurine and 2 mM carnitine.

### 4.4. Langendorff Isolated Heart Perfusion

Referring to previous research [13], a Krebs–Henseleit (K–H) buffer (in mM) was prepared: 131 NaCl, 4.0 KCl, 1.2 KH_2_PO_4_, 1.2 MgSO_4_, 1.8 CaCl_2_, 25.0 NaHCO_3_ and 11.0 glucose equilibrated with mixed gas (95% O_2_ and 5% CO_2_) to a pH of 7.35–7.45 at 37 °C. The rat was anesthetized by pentobarbital sodium (40 mg/kg) intraperitoneally and following intravenous injection of heparin (500 UI/kg), the heart was excised. After draining the residual blood with the K–H buffer, the heart was hung on the Langendorff perfusion model and subjected to retrograde perfusion with the K–H buffer at a constant pressure (80 mmHg). The tested chemical BTA07 was pre-perfused, followed by the ischemic perfusion and re-perfusion in the presence of BTA07 as described by Cai et al. [9]. In the process of experimentation, hemodynamic parameters (left ventricular end diastolic pressure, LVEDP; maximum rate of rise/fall of left ventricular pressure ±dP/dt max; heart rate, HR) were monitored and recorded by the Lab Chart system.

### 4.5. Experimental Groups and Treatments

The experimental groups and timeline are presented in Figure 2. Thirty rats were randomly divided into six groups (five per group) as follows: (1) normoxia group; (2) ischemia–reperfusion group (vehicle), in which the ischemia hearts with the glucose-free K–H buffer equilibrated with 95% N_2_ and 5% CO_2_ for 50 min followed by 120 min of reperfusion with the glucose K–H buffer; (3) BA 4.57 mg/L treatment group (BA 10 μM), in which the isolated hearts were perfused with the K–H buffer containing BA for 20 min after an equilibration period. Then, the hearts were perfused with the glucose-free K–H buffer equilibrated with 95% N_2_ and 5% CO_2_ for 50 min after stabilization, followed by 120 min of reperfusion with BA; (4) BTA07 0.60 mg/L treatment group (BTA07 1.1 μM), in which the isolated hearts were conducted as per (3); (5) BTA07 1.77 mg/L treatment group (BTA07 3.3 μM), in which the isolated hearts were conducted as per (3); and (6) BTA07 5.31 mg/L treatment group (BTA07 10 μM), in which the isolated hearts were conducted as per (3).

### 4.6. In Vitro Hypoxia/Reoxygenation Model

As previously described [40], in vitro experiments were used for freshly isolated adult rat cardiomyocytes (ARCMs). Then, cells were attached onto culture dishes pretreated with 1.0 μg/mL of laminin for 2 h. Cultured M199 medium without serum was replaced to ensure that the survival rate was more than 95%, and then cultivated for 4h for subsequent experiments under normal oxygen conditions. Afterwards, ARCMs were subjected to hypoxia/reoxygenation (H/R) to mimic the model of MIRI in vitro in the presence of BTA07. Briefly, ARCMs were treated with and without BTA07 (10 μM) in the presence or absence of LY294002 (10 μM). Then, ARCMs were maintained for 8 h in an oxygen-free incubator (95% N_2_ and 5% CO_2_) at 37 °C. Cells were subsequently incubated for 9 h or reoxygenation under an incubator (95% air and 5% CO_2_) at 37 °C. The normoxia control group was placed in 95% air and 5% CO_2_ for 17 h.

### 4.7. Measurement of Myocardial Infarct Size

Myocardial infarct size was measured by TTC staining as previously described [41]. In brief, the isolated hearts were removed from the device and frozen at −80 °C for 30 min. The heart was then sectioned with a thickness of 2 mm by a heart cutter. These pieces were evenly stained using 1% TTC, washed with PBS buffer and fixed using 10% formalin for 24 h. The infarct area and the normal area were white and brick red, respectively. Finally, photos were taken with a digital camera and the myocardial infarct area was calculated with Image J software version 1.8.0.112(NIH, Bethesda, MD, USA).

### 4.8. Histopathological Observation of Damaged Myocardium Tissue

At the end of the reperfusion, 1 mm^3^ of the left ventricle was immediately cut and fixed by 4% paraformaldehyde solution to prepare frozen sections. After dehydration and embedding in paraffin, the tissue was sliced into 5 µm-thick slices. These slices were stained by routine hematoxylin–eosin (H&E) and pathological changes in the myocardium were then observed under a light microscope.

### 4.9. Detection of Biochemical Indexes

The coronary outflow was collected within 120 min after reperfusion as a designed time point. Myocardial tissue enzyme lactate dehydrogenase (LDH) and creatine kinase-MB (CK-MB) levels of coronary outflow were detected to evaluate cardiac injury using commercial kits according to the manufacturer’s instructions. The LDH release in ARCMs was also measured with an LDH cytotoxicity detection kit. Then, oxidative stress marker levels for malondialdehyde (MDA), superoxide dismutase (SOD) and glutathione peroxidase (GSH-Px), as well as apoptosis-related factor caspase-3 activity in myocardial tissue homogenate (ex vivo) and ARCMs (in vitro), were measured by detection kits according to the manufacturer’s instructions.

### 4.10. Propidium Iodide Assay

A propidium iodide (PI) assay was used to quantify the death of cardiomyocytes. ARCMs were rinsed with a PBS buffer after H/R treatment and incubated with 10 μg/mL of Hoechst H33342 for 15–30 min at 37 °C for cell counting. The cells were then rinsed three times with PBS and incubated with 10 μg/mL of PI for 20–30 min at 37 °C for dead cell counting. A Leica fluorescence microscope was used to capture the images. The excitation and emission wavelengths for PI staining were 536 nm and 617 nm, respectively. The excitation and emission wavelengths for Hoechst staining were 350 nm and 461 nm, respectively.

### 4.11. Measurement of Reactive Oxygen Species

MitoSOX™ (Thermo Scientific) was used to detect the level of ROS in cells. Cell suspensions were obtained by tryptase digestion and 2′,7′-dichlorofluorescein was used as a probe. After administration, cardiomyocytes were incubated with MitoTracker Green (200 nM, Beyotime, Shanghai, China) and MitoSOX^TM^ (5 µM, Thermo Scientific, Waltham, MA, USA) at 37 °C in the dark for 30 min, followed by DAPI staining for 10 min and detection of fluorescence intensity using a laser confocal microscope. The excitation and emission wavelengths for MitoTracker Green are 490 nm and 516 nm, respectively. The excitation and emission wavelengths for MitoSOX^TM^ are 510 nm and 580 nm, respectively. The excitation and emission wavelengths for DAPI staining are 340 nm and 488 nm, respectively.

### 4.12. Western Blotting

Proteins were extracted from isolated hearts and ARCMs, determined by a BCA assay kit (Beyotime Biotechnology, Shanghai, China), separated by 12% sodium dodecyl sulfate–polyacrylamide gel electrophoresis (SDS-PAGE) and then blotted onto polyvinylidene difluoride (PVDF) membranes (Millipore, Billerica, MA, USA). The membranes were then probed with primary antibodies (1:1000) diluted in 5% nonfat milk and incubated at 4 °C overnight. After a washing step with tris-buffered saline and Tween 20 (TBS-T), membranes were incubated with horseradish-peroxidase-conjugated secondary antibodies for 2h at room temperature. Immune complexes were visualized by enhanced chemiluminescence after additional washing and band intensity was measured quantitatively with Image J software.

### 4.13. Statistical Analysis

The data are presented as mean ± S.E.M. from three or five independent experiments. Statistical analyses were performed using GraphPad Prism version 8.0.1 (IBM Corp., Armonk, NY, USA). Significant differences were analyzed using a two-way ANOVA test and an one-way ANOVA test followed by a Student–Newman–Keuls test. *p* < 0.05 was considered to indicate a statistically significant difference.

## 5. Conclusions

This study found that BTA07 significantly improved isolated heart injury induced by I/R, reduced myocardial infarction size, inhibited oxidative stress and regulated the activation of caspase-3. Accordingly, the protective effect of BTA07 in H/R-induced ARCM damage and apoptosis was at least partially mediated by the PI3K/Akt pathway. Therefore, this study lays a foundation for the further study of BTA07 preventing oxidative stress and myocardial apoptosis in MIRI, and provides a novel idea for the development of a novel anti-MIRI drug that is highly effective and has low toxicity by microbial transformation.

## Figures and Tables

**Figure 1 ijms-23-09473-f001:**
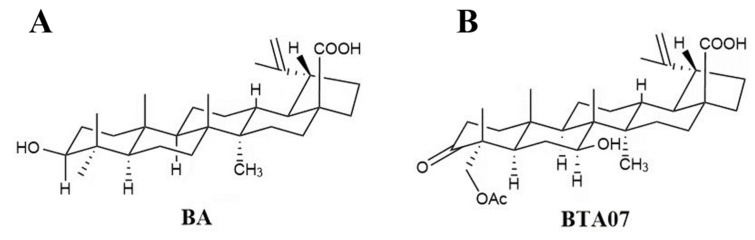
The structures of (**A**) betulinic acid and (**B**) BTA07.

**Figure 2 ijms-23-09473-f002:**
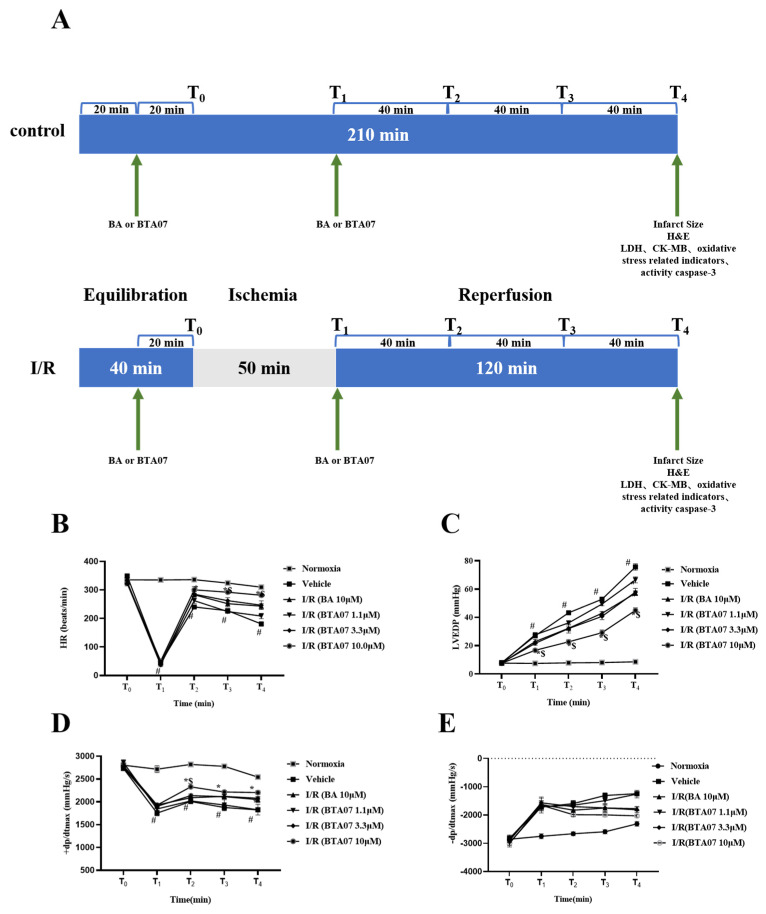
BTA07 treatment improved the hemodynamic parameters of isolated rat hearts. (**A**) Animal grouping and timeline ex vivo. Heart rate (HR, (**B**)) left ventricular end diastolic pressure (LVEDP, (**C**)), maximum rate of risen left ventricular pressure (+dP/dt max, (**D**)) and maximum rate of fallen ventricular pressure (−dP/dt max, (**E**)) in the isolated rat heart preparation were monitored and recorded by Lab Chart. Data were presented as means ± S.E.M for five independent experiments. ^#^ *p* < 0.05 vs. normoxia group; * *p* < 0.05 vs. vehicle group; ^$^ *p* < 0.05 vs. BA group; two–way ANOVA test, followed by a Student–Newman–Keuls test.

**Figure 3 ijms-23-09473-f003:**
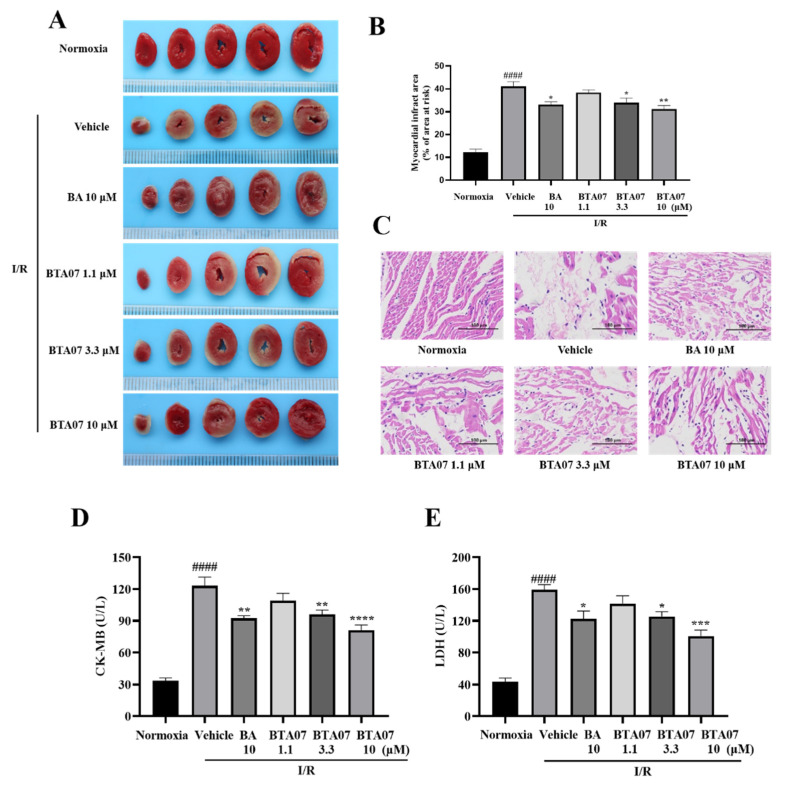
BTA07 ameliorates I/R-induced cardiac tissue damage in isolated rat hearts. (**A**) Representative images of myocardial infarct size by TTC staining. (**B**) Myocardial infarct sizes were presented as a percentage of the infarct area/total area. (**C**) Representative pictures of H&E-stained cardiac sections (*n* = 5 per group). Magnification 200×, scale bar = 100 μm. (**D**) Mean levels of lactate dehydrogenase (LDH) from the coronary outflow effluent in all groups. (**E**) Mean levels of creatine kinase-MB (CK-MB) from the coronary outflow effluent in all groups. Data are presented as means ± S.E.M for five independent experiments. ^####^ *p* < 0.0001 vs. normoxia group, **** *p* < 0.0001 vs. vehicle group; *** *p* < 0.001 vs. vehicle group; ** *p* < 0.01 vs. vehicle group; * *p* < 0.05 vs. vehicle group; one–way ANOVA test, followed by a Student–Newman–Keuls test.

**Figure 4 ijms-23-09473-f004:**
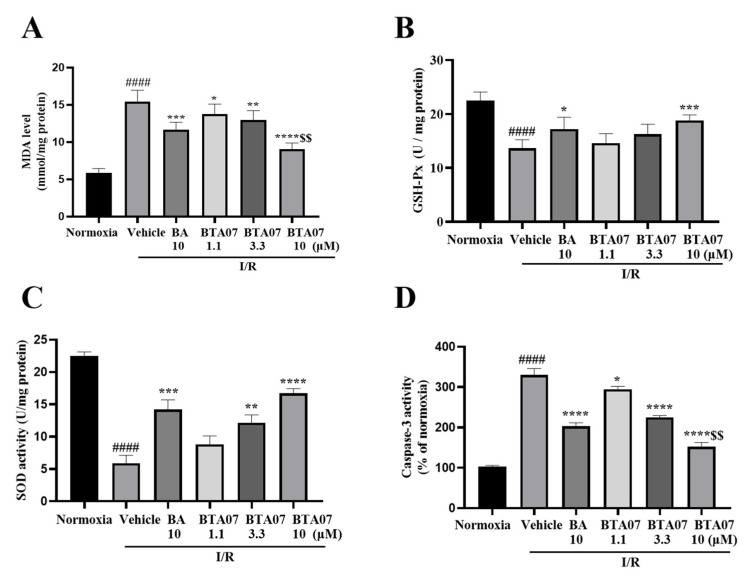
BTA07 treatment suppressed I/R−induced oxidative potential and apoptosis in cardiac tissue. (**A**) Levels of malondialdehyde (MDA). (**B**) Glutathione peroxidase (GSH-Px). (**C**) Superoxide dismutase (SOD). (**D**) Caspase-3 activity in the isolated rat heart preparation was detected in the myocardial homogenate of isolated hearts after reperfusion. Data are presented as means ± S.E.M for five independent experiments. ^####^ *p* < 0.0001 vs. normoxia group; **** *p* < 0.0001 vs. vehicle group; *** *p* < 0.001 vs. vehicle group; ** *p* < 0.01 vs. vehicle group; * *p* < 0.05 vs. vehicle group; ^$$^ *p* < 0.01 vs. BA group; one–way ANOVA test, followed by a Student–Newman–Keuls test.

**Figure 5 ijms-23-09473-f005:**
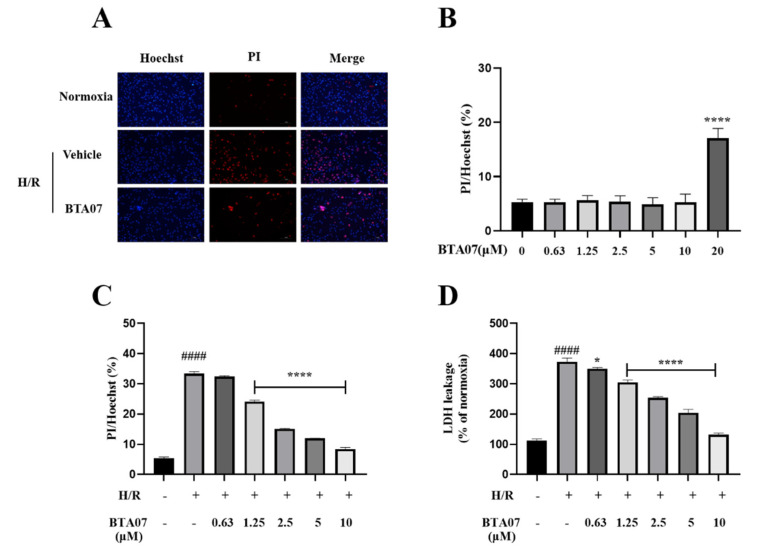
BTA07 decreased cell death rate and LDH release in ARCMs exposed to H/R. (**A**) Representative confocal microscopic images. (**B**) Cytotoxicity of BTA07 on ARCMs. The cells were incubated with different concentrations of BTA07 (0, 0.63, 1.25, 2.5, 5, 10 and 20 μM). (**C**) Effect of BTA07 on cell death rate induced by H/Rin ARCMs. PI/Hoechst staining was used to calculate cell death rate. (**D**) Effect of BTA07 on LDH release in H/R-induced ARCMs. Data are presented as means ± S.E.M for three independent experiments. ^####^ *p* < 0.0001 vs. normoxia group; **** *p* < 0.0001 vs. vehicle group; * *p* < 0.05 vs. vehicle group; one–way ANOVA test, followed by a Student–Newman–Keuls test.

**Figure 6 ijms-23-09473-f006:**
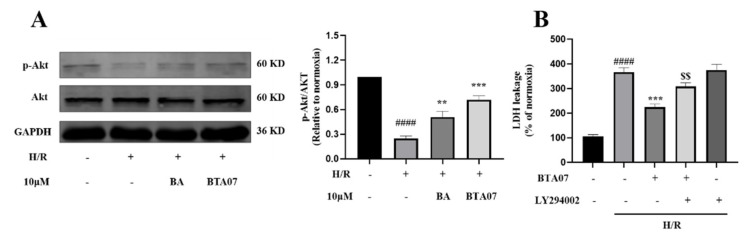
BTA07 recovered PI3K/Akt signaling suppressed by H/Rin ARCMs. (**A**) Left panel: a representative image; right panel: average data. (**B**) ARCM damage was determined by LDH release following treatment with 10 μM of LY294002. Data are presented as means ± S.E.M for three independent experiments. ^####^ *p* < 0.0001 vs. normoxia group; *** *p* < 0.001 vs. vehicle group; ** *p* < 0.01 vs. vehicle group; ^$$^ *p* < 0.01 vs. BA group; one–way ANOVA test, followed by a Student–Newman–Keuls test.

**Figure 7 ijms-23-09473-f007:**
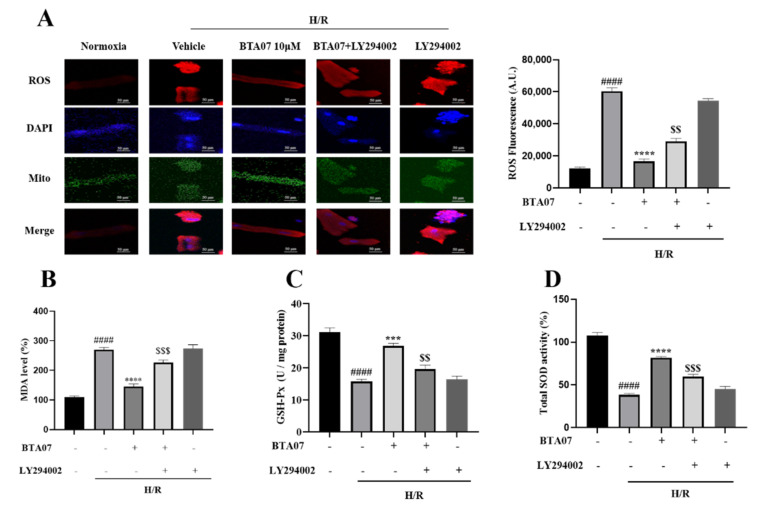
Pharmacological inhibition of PI3K with LY294002 abolished BTA07-induced antioxidative potential in ARCMs. (**A**) Representative images of a fluorescence probe for ROS in ARCMs. The concentration of ROS dye was 5 μM. Levels of malondialdehyde (MDA, **B**), glutathione peroxidase (GSH-Px, **C**) and superoxide dismutase (SOD, **D**) in the ARCM lysates were detected. Data are presented as means ± S.E.M for three independent experiments. ^####^ *p* < 0.0001 vs. normoxia group; **** *p* < 0.0001 vs. vehicle group; *** *p* < 0.001 vs. vehicle group; ^$$$^ *p* < 0.001 vs. BTA07 group; ^$$^ *p* < 0.01; one–way ANOVA test, followed by a Student–Newman–Keuls test.

**Figure 8 ijms-23-09473-f008:**
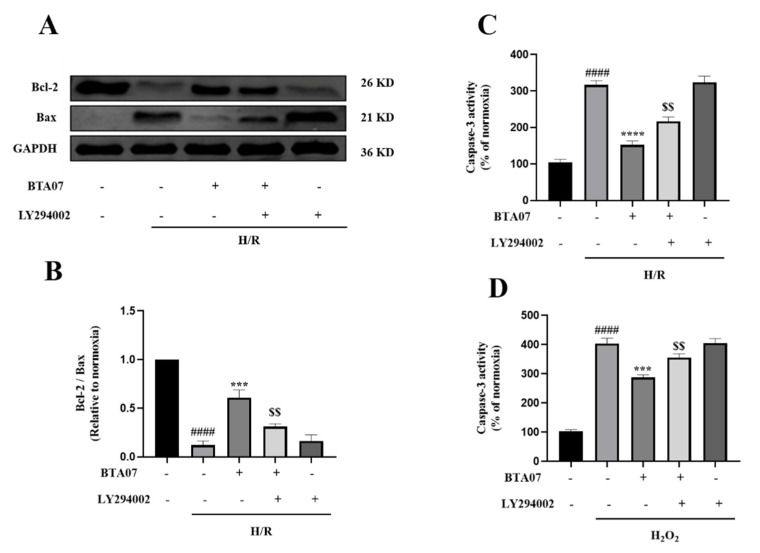
BTA07 inhibited H/R-induced apoptosis in ARCMs via the PI3K/Akt signaling pathway. (**A**,**B**) Expression of Bax and Bcl-2 were measured by Western blot. (**C**) Caspase-3 activity in ARCMs. (**D**) Effects of BTA07 on ARCMs caspase-3 activity after H_2_O_2_ was induced. Data are presented as means ± S.E.M for three independent experiments. ^####^ *p* < 0.0001 vs. normoxia group; **** *p* < 0.0001 vs. vehicle group; *** *p* < 0.001 vs. vehicle group; ^$$^ *p* < 0.01 vs. BTA07 group; one–way ANOVA test, followed by a Student–Newman–Keuls test.

## Data Availability

Data are available from the corresponding author upon request.

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
