# Peer review of "Cardiac Protection of a Novel Lupane-Type Triterpenoid from Injuries Induced by Hypoxia–Reperfusion"

_ijms, 2022, doi:10.3390/ijms23169473_

Round 1
Reviewer 1 Report
Dear authors, this paper is of high level and merit to be published in IJMS
My major concerns are on the short list of references, that could be improved and updated and on the option to include a Conclusion section.
Also is very important that the authors give the exact number of animals used and also more information and citations on the methods. The reference of each assay should be cited.
In material and methods the authors should report the details of the reagents’ suppliers
Please avoid to repeat terms in the title as keywords
line 40 and more: The term triterpenoid is sometimes erroneously typed as “triterpenoid”, please revise.
Typos at 237, please check and revise
Lines 290-298 check for editing
Line 383 Image J software, please include the suppliers and the version used
I would include a conclusion section
Also the MS is plenty of editing typos, such as CO2, N2, etc etc.
Reviewer 2 Report
The authors demonstrated that the application of a drug BTA07, a metabolite of Betulinic acid (BA) is able to ameliorate and to offer the protection against IRI in Langendorff hearts and isolated rat cardiomyocytes model of hypoxia/reoxygenation. This effect is lost in presence of PI3K/AKT inhibitor LY294002 indicating that the BTA07/phosphoAKT axis is the main BTA07-sensitive intracellular element in IRI. These molecular events are followed by the alteration in the Bax-Bcl2 balance that changes and governs the terminal intracellular signal such is the caspase activity.
Main concerns:
The experimental setup in the Langendorff model system is not so clear, neither it is explained why the treatment was performed using the actual pre-treatment with BTA07? Hearts are actually equilibrated with BTA07, then they are perfused with ischemic buffer (is all the BTA07 was washed off instead priming the tissue or the flow of the ischemic buffer was actually stopped?) and then the hearts were perfused with BTA07 again but with the presence of glucose to drive the production of ROS.
The ROS-mediated damage occurs mostly in the first 5 minutes of reperfusion, therefore it might be more important to treat the hearts for 5 min before the end of ischemia and then for additional period during the reperfusion, if the authors are keen to examine any of the antioxidative properties of BTA07. Additionally, accessing the levels of succinate and ATP/ADP ratio as a common parameter in IRI from ischemic and reperfusion phase might help elucidating the molecular mechanism.
It also seems that the cell experiments have the different treatment setup compared to the Langendorff model.
Authors are strongly advised to explore the current literature in respect of the difference between the Langendorff model of IR injury and anoxia/reoxygenation rather than hypoxia/ reoxygenation for reaching the better and physiologically more relevant comparison.
MAIN COMENTS:
There are many grammatical mistakes, wrong sentence structures and often spacing mistakes.
A detailed description of the cell hypoxia and reoxygenation experiments is missing.
A detailed explanation of PI3K/AKT and its role in H/R is missing (some are in discussion
and could be moved earlier, but the exact role in MIRI is unclear).
The link between LY294002 and BTA07 is not proven just hypothesised. It could be a correlation. Would a caspase inhibitor have the same anti-apoptotic effect as BTA07? Would another antioxidant have similar effects?
While apoptosis is investigated, other cell death pathways are not discussed at all.
The schematic overview in Figure 9 does not add anything as it just shows that the link between BAT07 and cardioprotection is not known/understood.
How does BAT07 works in the cell? Does it react/scavenge ROS directly or is the antioxidant effect the secondary effect?
The methods section absolutely needs more detailed descriptions quantification details.
DETAILED COMENTS:
Sentence 15: ‘trasmformation’ is misspelled word.
Sentence 49: The verbal construct ‘family of signal transduction’ is not common and authors are advised to correct this.
Introduce PI3K/AKT pathway and its role in MIRI better!
Sentence 91: Authors are advised considering changing this statement: gold standard
Figure 3: A-B) Infarct locations are not very clear, maybe it is worth to highlight them in the picture and to indicate what part was quantified? 3C) The low concentration of BTA07 seems to rescue cells already. How come that the heart parameters in Figure 2 were not that strikingly different? 3D-E) The figure legends is missing the information regarding the origin of analyte (e.g., LDH and CK-MB from the coronary outflow effluent/perfusate).
Sentence 120: Reduce, not inhibit the levels of MDA.
Sentence 124: Significantly or obviously?
Figure 4: Legend should be better describing/more details. Cell death assays (caspase and LDH) should have a positive control.
Sentence 148: ‘Insulting’ is the wrong word.
Figure 6A and 6B can be placed in the supplemental material is possible?
Maybe to combine Figures 5 and 6 in one?
Figure 7: The labels of ROS with which ROS probe is missing. How are the mitochondria stained? This figure requires a detailed description. It is not clear what is written in the figure legend and how this correlates with the graphs. Is it the levels of expression (levels of MDA, which is a small molecule?) or activity of enzymes such are SOD and GSH-Px. The figure is missing the word ‘total SOD’ since that is what the authors were using in the material and method section. The scale bar is missing on microscopy images. How was the quantification performed?
195: The BAX -BCL2 axis requires a brief explanation/introduction to help a reader.
205: Authors are advised to change the ROS-induced into ROS-mediated.
Sentence 209: ‘Somehow' is inappropriate term.
Figure 8: Using the exogenous H2O2 directly in cell culture is not always the best method to address the certain questions. In the case of studying the H/R the better way would be to elicit the mitochondrial ROS production rather than adding the exogenous one. However, chemical approach would be to use paraquat. In addition, it might be useful to include the positive control in caspase experiments.
Figure 9: The figure legend should include a description of the schematic. The figure does not show any connection between the drug and PI3K nor how this would be cardioprotective. It also does not include any explanation how it is related to H/R.
Why does the GAPDH look different in Figure 6 than in Figure 8?
Section 5.6 is poorly explained and require more detailed description. Which medium was used and what was its composition during hypoxia and reoxygenation.
Section 2.4 Authors are claiming that LY294002 affect the phosphorylation of PIP3 and Akt. Did the authors demonstrate this in their experiments? It seems that the authors are missing the data in Figure 6 that demonstrate the actual abolishing/competing effect of BTA07 on LY294002-induced inhibition of Akt or PIP3 phosphorylation, since the author statements is that BTA07 treatment significantly up-regulated the expression levels of p-Akt in ARCMs. The results demonstrating the LDH release, are far away as a way from a system used to demonstrate the indirect effect of abolishing/competing and proving of BTA07 towards phosphorylation of PIP3 or Akt.
Sentence 160: To clarity, or to clarify?
Sentence 160, 161 and 161: Authors are advised to introduce the changes as followed: levels of phosphorylation of Akt instead the expression of p-Akt.
Section 2.5
In the section 2.5 the experimental details regarding the results demonstrated in the Figure 7 are poorly explained in the figure legend. In addition, in the experimental section 5.6 ,authors used pre-treatment of cells with BTA07, then they performed BTA07-free ischemia followed by BTA07-free reperfusion medium. This experiment that actually supposed to mimic the in vitro hypoxia/reoxygenation model was not performed with the similar conditions the Langendorff experiments were, where the reperfusion medium contained the BTA07.
Is there any particular reason for introduction of this experimental modification?
Sentence 180: misspelled word: significantlyincreased
Sentence 183: The term ‘antioxidation’ is very wrong. Authors are strongly advised to change this.
The authors conclusion:
The significant amount data in conclusion that reflects the history, and the nomenclature of PTs molecule can be integrated in the introduction perhaps?
Sentence 229: This study does not explain the regulatory effect of BTA07 on the PI3K/Akt signalling pathway following MIRI, it rather provides the observation at this stage.
Sentences 232-237 are not essential, and authors are advise to remove them.
Sentenc3 256: Authors are advised to introduce them self with the in deep knowledge of the cellular model systems for studying the Ischemia-reperfusion. The partial pressure of Oxygen that can be achieved using hypoxic incubator is not even close to the one that can be achieved by using the Anoxic chamber. This is the key point in studying the Ischemia reperfusion like effect in cell culture since the biochemical and metabolic hallmarks of the Langendorff model are almost fully replicated using the Anoxic chamber, which is not the case by using the Hypoxic incubator.
See the following: https://doi.org/10.1016/j.redox.2022.102368
Authors are advised to introduce the chemical properties and the importance of the functional groups in the molecule of BTA07 and the Betulinic acid. Presence of the unsaturated carbon-carbon bond as well as the presence of the carbonyl group within the structure of the molecule might and most likely will affect its electrophilic nature. This later was demonstrated using the Keap1/Nrf2 system to test the electrophilic/antioxidative potential of Betulinic acid in vitro in cell model of hypoxia/reoxygenation. Therefore, authors are advised to comment weather the Nrf2-mediated effect of BTA07 might play an important role in cellular protection against IRI, in particular in the light of the experimental setup used by authors such as the BTA07 pre-treatment prior actual IRI.
Interestingly, the recent publication indicates that the Betulinic acid excrete the anticarcinogenic properties via PIP3/Akt pathway ultimately inducing apoptosis in several cellular model of cancer. It might be worth to make notation about this in the text.
Section 5.11 is poorly explained and require more detailed description (how much MitoSox was used? Was it subjected in hypoxia?) and many more details are missing.
Sentence 407: Poor description of the microscopy settings and following parameters.
Reviewer 3 Report
Guo et al. investigated the effects of a betulinic acid-derivated compound (i.e., BTA07) on myocardial ischemia/reperfusion injury in rats and cardiomyocytes. Their main finding is that BTA07 had a dose-dependent cardioprotective effect partly via inhibiting oxidative stress and apoptosis. The study seems to be carefully designed and evaluated using plenty of methods. I have several suggestions and questions to the authors.
1. Please give the ethical license number in the methods section.
2. How many rats were used in this study? Please give the total number and the number of animals in each group.
3. Only male rats were used in this study. Please mention this fact in the limitation section and discuss the potential sex-based differences in the effects of BTA07.
4. Why did the authors use glucose-free Krebs-Henseleit solution if they applied global ischemia to induce AMI? Please discuss it.
5. Minor comment: in many cases, spaces are missing between the words. Please read your text carefully or use spell-checking software to correct this mistake.
Round 2
Reviewer 2 Report
The main comment to authors mostly based on questions: 2, 3, 5, 7, 12 (Fig7):
It is not clear weather authors understand the importance of having the same experimental conditions even between different model systems. The ex vitro experiments were performed within 210 (3.5h) minutes compared to the isolated rad cardiomyocyte in vitro model system that was performed within 17 hours and they had a different treatments and pre-treatments with BTA07.
However, the in sight into molecular mechanism of BTA07 in favour of caspase inhibition was obtained mostly from the cell model, whereas demonstrated in Figure 7, BTA07 applied alone (when was the ROS probe added, after reperfusion or before and why the original micrographs were replaced by the other ones?)
In the Langendorff model from where the only indirect ROS metabolite MDA was measured, the effect of BTA07 on acute ROS production might be more important since the data were obtained in the significantly shorter time frame after IRI.
Therefore, In the Langendorff experiments, it is not so clear from which phase of the BTA07 treatment (priming or perfusion) the beneficial effects are coming from?
The effect on caspase activation might also originate from the rapture of the mitochondrial membrane leading to the loss of cytochrome c, the main apoptosome assembly factor. Somehow, the authors were able to perform but, not to comment the experiment that links exactly this effect by demonstrating the lower levels of MDA, the main membrane lipid peroxidation metabolite in the BTA07 treatment which accumulation is indicative for the rapture of the membrane integrity and the loss of cytochrome c.
Authors are seriously advised to arrange the English language proof reading services. The current version of the manuscript requires the professional writing-editing correction in order to reach the higher quality and the originality.
Round 3
Reviewer 2 Report
The authors manage to address the reviewers question in their final rebuttal letter with more sense than in the previous versions of their manuscripts.
I do hope that this review proves was helpful for authors in particular if they have any further aspiration towards continuing to explore the BTA07 in their future studies.
Authors are advised to check the final proof in detail introspect to misspelling and similar grammatical mistakes.
Author Response
Q1: The authors manage to address the reviewers question in their final rebuttal letter with more sense than in the previous versions of their manuscripts. I do hope that this review proves was helpful for authors in particular if they have any further aspiration towards continuing to explore the BTA07 in their future studies.
A1: Thank you so much for constructive comments and stimulating idea. All of the suggestion and comments will definitively help us to move the BTA07 project forward. In fact, our lab will make more compounds based on the BTA07 and further explore the molecular mechanism protected hearts from the injuries induced by the ischemia. At the same time, your comments have indeed improved the manuscript quality and originality. All of authors appreciated you and your advices.
Q2: Authors are advised to check the final proof in detail introspect to misspelling and similar grammatical mistakes.
A2 : Yes. To ensure the language original, the revised manuscript is edited by professional one (english-48937).

This manuscript is a resubmission of an earlier submission. The following is a list of the peer review reports and author responses from that submission.
Round 1
Reviewer 1 Report
This is an experimental study on the in vitro and ex-vivo cardio-protective effects of the BA derivative BTA07 from myocardial ischemia-reperfusion injury.
The study design is adequate, the results clearly presented, the discussion exhaustive.
I believe that a reorganization of the paragraphs is necessary to make the text clearer and conform to the editorial rules of scientific work.
In particular:
- section 4 Materials and Methods with its paragraphs must become section 2;
- lines 272-282 anticipate some conclusions and describe the limits of the study: I believe that the limits should be highlighted in a special section, placed after the discussion, and that the conclusions should be cited as such only in the conclusions section.
I also suggest a minor check of the English language.

Author Response
Q1:“I believe that a reorganization of the paragraphs is necessary to make the text clearer and conform to the editorial rules of scientific work.In particular:-section 4 Materials and Methods with its paragraphs must become section 2;”
A1:Many thanks. According to your advice, the section of Materials and Methods was replaced in the revised manuscript. However, our original manuscriptwasformattedbased on the guidelines of the IJMS. Nevertheless, we believe the editorialteamwillmodify this formatto adhere to their guidelines.
Q2: “-lines 272-282 anticipate some conclusions and describe the limits of the study: I believe that the limits should be highlighted in a special section, placed after the discussion, and that the conclusions should be cited as such only in the conclusions section.I also suggest a minor checkof the English language.”
A2: A great point. According to your advice, the limitations and conclusions were moved to specific sections. A native English speaker carefully read the revised manuscript.

Reviewer 2 Report
This study investigates protective effects of BTA07 in reperfusion injury based on experiments with isolated rat hearts and adult rat ventricular cardiomyocytes. The drug is given prior to ischemia.
Major comment:
The main problem is that the intention of the study is not clear. Is it to compare BA with BTA07 asd the latter one can be better solved in water? Or is it to identify the mechanism of BA by replacing BA with BTA07 for any reason? Furthermore, the first sentence of the abtract (Background) is unclear: MIRI can lead to myocardial infraction? How do you define myocardial infarction? Do you mean to post-ischemic heart failure?
Don not use abbreviations like ARCMs in abstract!
You state that you have confirmed the protective effect of BTA07 (this was known?!). Or do you mean BA?
X-Axis in Fig. 2 can either be Time in h as illustrated or the different time points as suggested by 2A.
Y-Axis in Fig. 2: The unit for dp/dt is wrong. This cannot be mmHg!
Please add basal and final values for LVDP.
Why is the concentration for BA and BTA07 identical, if one of them can be better solved in water? Again, do you want to compare both drugs? If yes, why then not compare this in the cell model?
Fig. 5: Submit light microscopy from the myocytes so that we can see whether the myocytes are rod-shaped (protected) or rounded (damaged).
Provide a WB with more than one sample per group (for each of the three independent experiments).
Why should isolated cells release LDH? By hypercontraction they should be intact and you talk about apoptosis not necrosis, so there should be no membrane damage. It seems that the membrane gets fragile but no calcium overload occurs as in the isolated heart model. Difficult to explain the mechanism of one experiment by another if the cells behave so different.
Fig. 6: Does Ly block Akt phosphorylation? That’s important for the conclusion but the experiment is missing!
In Fig. 7 we see that all myocytes are still rod-shaped. What does this mean for the simulated H/R model?
The meaning of the LY experiments is unclear. Why less ROS without Ly?
Fig. 9 unclear. What does the drug do? If P-Akt is acting via bax and bcl2 expression (as suggested by your WBs) than the arrow from P-Akt has to go to the nucleus not the mitochondria! Why does bax/bcl2- limit ROS production? How did you exclude that BTA07 affects the expression of anti-oxidative pathways like SOD and that then less ROS triggers the expression of bax/bcl2 instead of reducing the levels of ROS by downregulation of proteins that control a mega-pore?
What about coronary flow. Affected by the drug? Pressure constant perfusion means that some hearts might be under-perfused during reperfusion. Add coronary resistance.
It reads as all hearts were treated with BA during reperfusion. “in which the isolated hearts were conducted as (3). True? If yes, why did you replace BTA07 (as give pre-ischemic) during reperfusion?
If the non-beating ARCM have still M199 in the hypoxic chamber there should be enough ATP via glycolysis to keep then non-overloaded by calcium. To provoke them you must remove glucose. This explains the rod-shaped cells after reoxygenation but then it does not mimic the Langendorff experiments.
Please give absolute activity for the enzyme measurements (such as U/mg protein in all figures including Fig. 7) and add the detection limit of the test kits in the material section. Please add the time point at which you collected the coronary outflow.
Please add which variables you used for the two-way ANOVA and indicate such experiments in the figure legend. Please replace Student’s T-test by the more robust Student-Newman-Keuls test (as posthoc) or Bonferroni with correction for multiple testing.
Please replace the term ‘suppressed expression of p-Akt’ by suppressed phosphorylation and thereby activity of p-Akt. That is what you measure.
In the isolated cell model, BRA07 almost completely protected against cell death. However, functional recovery was improved but not normalized to normoxic controls. Why?
Author Response
Reponses to Review2
Q1: “The main problem is that the intention of the study is not clear. Is it to compare BA with BTA07 and the latter one can be better solved in water? Or is it to identify the mechanism of BA by replacing BA with BTA07 for any reason?”
A1: Thank you so much for your comments. Indeed, there is a logical issue in the original abstract. The abstract and introduction were edited in the revised version:
“Since ischemic heart disease is one of the leading causes of death worldwide, it is urgent to find effective drugs for myocardial ischemia from natural resources. Betulinic acid (BA), a widely distributed lupane-type triterpenoid, has been reported to possess anti-oxidative activity and inhibit apoptosis against ischemic heart disease. Because of low bioavailability and water insolubility of BA, previous studies havecreateda series of BA-derivative compounds by microbial transformation, but it is unclear of the underlying mechanisms of these compounds and whether theyhave cardiac protection properties. Here, we observe the anti-MIRI effects of BTA07, a BA derivative, on the injuries induced by hypoxia/reoxygenation (H/R) in adult rat cardiomyocytes (in vitro) and in Langendorff-perfused heart ex vivo. We will further explore its mechanism of cardioprotection.”
Q2:“Furthermore, the first sentence of the abstract (Background) is unclear: MIRI can lead to myocardial infraction? How do you define myocardial infarction? Do you mean to post-ischemic heart failure?”“Don not use abbreviations like ARCMs in abstract!”
A2: According to your advice, the term “adult rat cardiomyocytes (ARCMs)” was added in the revised manuscript. Yes, the first sentence is logically misleading. It is corrected in the revised version.
Q3 “X-Axis in Fig. 2 can either be Time in h as illustrated or the different time points as suggested by 2A.
Y-Axis in Fig. 2: The unit for dp/dt is wrong. This cannot be mmHg!
Please add basal and final values for LVDP.”
A3: We apologize for the mistakes. Here, “h” was changed to“min”, and “mmHg” was changed to “mmHg/s” in Fig. 2. In addition, the basal and final values for LVEDP were added in the revised manuscript. As we understand, the basal and final LVDP are displayed at To and T4 points, respectively, as shown in Figure2C.
Q4: “Why is the concentration for BA and BTA07 identical, if one of them can be better solved in water? Again, do you want to compare both drugs? If yes, why then not compare this in the cell model?”
A4: We believe that it is more convincing to compare the protective effects of the two drugs at the same concentration. Here, BA is only used as the positive control since it is a substrate for microbial transformation.
About water solubility, the two drugs at 10 μM are well dissolved in water. However, comparing the structure and the separation process of the two compounds, it has been found that BTA07 has greater polarity (Reference 3 in the revised manuscript), and it implied that it is more water-soluble, but this scientific question is not our focus in the present study. At the level of isolated heart ex vivo, BTA07, as its substrate BA, has been proved to have potential in anti-ischemic effects, evidenced by reducing the area of myocardial infarction and diminishing oxidative stress and apoptosis. Furthermore, the protective mechanism is to be defined at the cellular level. Herethe BA is considered as a positive control as shown in Figure 6A.
BA has shown cardiac protection as well (seen below in A for review). This data is not displayed in the manuscript since the model of cultured cardiac myocytes was to define the mechanism of cardiac protection.
Q5: “Fig. 5: Submit light microscopy from the myocytes so that we can see whether the myocytes are rod-shaped (protected) or rounded (damaged).
Provide a WB with more than one sample per group (for each of the three independent experiments).”
A5: In general, the rod shape cardiomyocytes are alive and the rounded onesaredamaged. However, in a real experiment, the rounded ones arenot all damaged and rod-shaped cells are not all alive, evidenced by some of the rod-shaped cells were positive with propidium iodide (PI) staining while some of the rounded cells were negative with propidium iodide (PI) staining. Thus, based on the principle of staining with propidium iodide (PI) and Hoechst, the PI positive impliesthat thatmembrane is damaged while Hoechst staining positive just displays cell counts. Thus, the representative images of Hoechst and PI staining were only displayed in the results section.
As requested, here are raw images in Figure 6A and Figure 8A, respectively.
Figure 6A
Figure 8A
Q6: “Why should isolated cells release LDH? By hypercontraction they should be intact and you talk about apoptosis not necrosis, so there should be no membrane damage. It seems that the membrane gets fragile but no calcium overload occurs as in the isolated heart model. Difficult to explain the mechanism of one experiment by another if the cells behave so different.”
A6: Thank you for this thoughtful question. LDH is the key enzyme for the biological synthesis of lactic acid, which mainly exists in myocardial tissue.
Toselect indicators or readouts in any experiments, we should consider the experimental time and experimental conditions to explore suitable detection methods based on the scientific question to be answered. The disorder of cardiac energy metabolism is an initial evoker to myocardial cell injury, and it is an important factor that causes and promotes the cardiac dysfunction. Therefore, indicator-related energy metabolism is often used as a readout of myocardial injury and drug evaluation. The lack of oxygen will change the capacity of metabolism in mitochondria from aerobic oxidation to anaerobic fermentation. At the same time, a large amount of lactic acid will be produced, resulting in intracellular acidosis. When the myocardial cells are damaged or necrotic, the level of LDH will be increased in the supernatant or the perfused out-flow. On the other hand, LDH is an important and specific marker for clinical diagnosis of myocardial cell damage.
Q7: “Fig. 6: Does Ly block Akt phosphorylation? That’s important for the conclusion but the experiment is missing!”
A7: Many thanks for the insightful comment. It is well known that LY used in the present study is a specific blocker for PI3K, an upstream kinase of AKT. The information is inserted into the context in the revised version. Although there is no data about the LY compound suppressing the phosphorylation of AKT in the present study, the LY compound indeed functionally masks the cardioprotection of BTA07, implying the LY-sensitive pathway is involved in the cardioprotection of BTA07.
Q8: In Fig. 7 we see that all myocytes are still rod-shaped. What does this mean for the simulated H/R model?
A8:In fact, the percentage of PI staining positive is about 25-35%.That means 65-75% of cells are alive, so their cellular membrane is intact or not damaged. Although most of thecells appearrod-shaped, the change incellular shape andmuscles fibers arranged disorderly with unclear transverse striationwere observed under the light microscope.
Q9: The meaning of the LY experiments is unclear. Why less ROS without Ly?
A9: As presented in Figure 6A, the protein levels of p-Akt were significantly suppressed by H/R in ARCMs compared to the normoxia group. More importantly, BTA07 treatment significantly upregulated the phosphorylation of Akt in ARCMs.It is well known that AKT is phosphorylated by its upstream kinase PI3K, implying that PI3K is involved in the BAT07-induced cardioprotection. Thus, todetermine the cause and effect relationship between ROS production and LY-sensitive target PI3K/AKT signaling, the levels of ROS evoked by hypoxia-reoxygenation (H/R)were measured in the absence and presenceof LY294002. As expected, the levels of ROS were increased in H/R and reduced by the BTA07 treatment. More interestingly, BTA07-caused less ROS was enhanced by LY294002, implying that LY294002 sensitive signaling is involved in the cardioprotection of BTA07.
Q10: Fig. 9 unclear. What does the drug do? If P-Akt is acting via bax and bcl2 expression (as suggested by your WBs) than the arrow from P-Akt has to go to the nucleus not the mitochondria! Why does bax/bcl2- limit ROS production? How did you exclude that BTA07 affects the expression of anti-oxidative pathways like SOD and that then less ROS triggers the expression of bax/bcl2 instead of reducing the levels of ROS by downregulation of proteins that control a mega-pore?
A: Thank you so much for your insightful and constructive comments. It is indeed unclear how BTA07 modulates the PI3K/AKT signaling and consequently regulates the BAX/Bcl2 and ROS signaling. The schematic in Figure 9 briefly summarizesthe results and draws the conclusion. Based on all results present in the manuscript, we proposed that BTA07 targets the PIK3/AKT signaling pathway. AKT was perhaps translocated into the mitochondria, in which it modulates ROS production or into nuclear to regulate the expression of Bax/Bcl2. Another possibility is that mitochondria-original ROS regulates BAX/Bcl2 expression by mitochondria-nuclear couple signaling. As you proposed,BTA07 might affect the expression of anti-oxidative pathways like SOD and consequently, less ROS triggers the expression of Bax/bcl2. Thus, it is worth further studying the cause-effect relationship between ROS-produced signaling and Bax/Bcl2, although both are regulated by the PI3K/AKT signal pathway as the present study proposed. These points are discussed in the first paragraph on page 11.
Q11: What about coronary flow. Affected by the drug? Pressure constant perfusion means that some hearts might be under-perfused during reperfusion. Add coronary resistance.
A11: The system used in the present study is the pressure constant perfusion. Since the present study focused on the cardioprotection of the tested compounds, unfortunately, the vessel resistance of the coronary system, evidenced by coronary flow, has not been observed. It is worth further exploring if BTA07 affects the vascular resistance when protecting the heart from injuries induced by H/R. This point is added in the discussion section (page 11, last sentence in second paragraph) . However, in the contraction function experiment of isolated rat mesenteric resistant vessels, BTA07 did not induce the vessel contraction and did not relax the pre-contractile vessel.
Q12: It reads as all hearts were treated with BA during reperfusion. “in which the isolated hearts were conducted as (3). True? If yes, why did you replace BTA07 (as give pre-ischemic) during reperfusion?
A: The ex vivo study has 6 groups as described in the Methods section. They are the Normia, H/R,H/R+BA (10 mM),H/R+BTA07( 10, 3.3, 1.1mM), respectively. The study is to test the cardiac protection of the tested drugs in the same experimental protocol.
Q13: If the non-beating ARCM have still M199 in the hypoxic chamber there should be enough ATP via glycolysis to keep then non-overloaded by calcium. To provoke them you must remove glucose. This explains the rod-shaped cells after reoxygenation but then it does not mimic the Langendorff experiments.
A13: These are great points. Many thanks. The rod-shaped cells were observed in the hypoxic chamber after reoxygenation since it might be non-overloaded by calcium, but it does not mimic the Langendorff experiments. In fact, some of the cardiomyocytes were indeed damaged in the present culture condition with M199 containing glucose, evidenced by PI staining positive.
Q14: Please give absolute activity for the enzyme measurements (such as U/mg protein in all figures including Fig. 7) and add the detection limit of the test kits in the material section. Please add the time point at which you collected the coronary outflow.
A14: Since the enzyme content of the cardiomyocytes isolated from different batches were significantly different, and the ratio to the control group was still statistically significant, some of the changes in enzyme activity in Figure 7 were present with the ratio.
It is indicated that the perfusate shall be collected for detection 120min after reperfusion in Fig.2A, and we also displayed the time points in methods 2.9 intherevised manuscript.
Q15: “Please add which variables you used for the two-way ANOVA and indicate such experiments in the figure legend. Please replace Student’s T-test by the more robust Student-Newman-Keuls test (as posthoc) or Bonferroni with correction for multiple testing.”
A15: Many thanks, all of these points were modified in the revised version.
Q16: “Please replace the term ‘suppressed expression of p-Akt’ by suppressedphosphorylation and thereby activity of p-Akt. That is what you measure.”
A16: According to your advice, the term was replaced in the revised manuscript. Many thanks.
Q17:“In the isolated cell model, BRA07 almost completely protected against cell death. However, functional recovery was improved but not normalized to normoxic controls. Why?”
A17: This is an insightful question. We believe that the different potentials of the tested drugs in the different experimental models are a popular biological phenomenon. More importantly, both models used in the present study displayed the cardiac protection of BTA07, although the contractile function ex vivo is not FULLY recovered by a tested compound BTA07.
It is an established concept that conditions in cell culture are simple and easy controlled, but the conditions at the organ level especially ex vivo are complex. The current research results suggest that BTA07 has greatly improved the survival of the cultured cardiomyocyte, but the cellular function including contraction and Ca2+ transient may be somehow damaged or at least not all of the cardiomyocytes have been improved in contractile contraction. At the organ level, a variety of factors work together to keep the cardiac contraction normal. In the present experiments, based on the basal heart rate and contractility of the perfused heart shown in Figure 2, the isolated heart worked well. After the perfusion with BTA07, the cardiac injury induced by ischemia reperfusion has been improved significantly, just like the results of cell culture, but the function did not recover completely. These results suggest that similar conclusions can be drawn from cell culture in vitro and the working heart ex vivo in terms of cardiac tissue injury. It is because of the similarities and differences in the results obtained in each model that the present data provide the promise to further study if BTA07 can be used to protect against ischemia-reperfusion injury by cardiac infarction in vivo.

Round 2
Reviewer 2 Report
I thank the authors for improvement of their study. However, as the aim of the study was to use the drug to clarify mechanistic pathways rather than to demonstrate again that the drug is protective, I guess that this is not really done. The final Figure still shows a couple of events that the authors have analyzed but no real mechanistic analysis as we do not know how these thinks are linked. Specifically, apoptosis was measured in cells indirectly but it can hardly affect the Langendorff readout as the time-sequence does not fit. In addition, the discussion about LDH and energy consumption is interesting, but here LDH is released from ischemic hearts by disruption of cells (necrosis) whereas the isolated cells are NOT necrotic. They may be damaged in any way but the one does not explain the other. In you state that these cells release LDH to scope with lactate production they first should not run into energy deficiency (why are there damaged in this case?) or alternatively no other more clinical relevant damage parameters such as TnI must NOT increase (but this is not shown). In any way this is a strong difference between the two models you use and in addition with the complete different time-schedule it is not possible to explain the one with the other.
In the Langendorff experiment the authors may either used a constant-flow model and can easily calculate coronary resistance by perfusion pressure that must be recorded in the system to avoid edema, or used a pressure-constant model and can then easily calculate coronary resistant by measuring the flow. In any kind, if they use a pressure constant model an coronary resistance increases the have a stronger mechanic stress to the endothelium that may affect the readout. If the used a constant flow then an increase in coronary resistance means that the different readout may be affected by different oxygen supply. As the drug acts on Akt-Pathway that is also in endothelial cells active this is a critical point that needs to be addressed. The authors cannot simply say that they did not measure this. This is an important part of the study protocol.
Author Response
Q1: I thank the authors for improvement of their study. However, as the aim of the study was to use the drug to clarify mechanistic pathways rather than to demonstrate again that the drug is protective, I guess that this is not really done. The final Figure still shows a couple of events that the authors have analyzed but no real mechanistic analysis as we do not know how these thinks are linked. Specifically, apoptosis was measured in cells indirectly but it can hardly affect the Langendorff readout as the time-sequence does not fit. In addition, the discussion about LDH and energy consumption is interesting, but here LDH is released from ischemic hearts by disruption of cells (necrosis) whereas the isolated cells are NOT necrotic. They may be damaged in any way but the one does not explain the other. In you state that these cells release LDH to scope with lactate production they first should not run into energy deficiency (why are there damaged in this case?) or alternatively no other more clinical relevant damage parameters such as TnI must NOT increase (but this is not shown). In any way this is a strong difference between the two models you use and in addition with the complete different time-schedule it is not possible to explain the one with the other.
A1: Thank you so much for your comments. We totally agreed with your points. It is emphasized in discussion that it is worth further studying the cause-effect relationship between ROS-produced signaling and Bax/Bcl2, although both are regulated by the PI3K/AKT signal pathway. Figure 9 briefly summarize the results and displayed the POSSIBLE relationship between PIK3/AKT, ROS, Bax/Bcl2 in cardiac myocytes. If you agree, Figure 9 could be not displayed in the manuscript. At the same time, the conclusion is changed to that BTA07 protects ARCMs and isolated hearts from hyopoxia-reperfusion partly through PI3K/AKT signal inhibiting oxidative stress and cardiomyocyte apoptosis in last sentence in abstract. “PI3K/AKT signal” is deleted because there is no any data about PI3K inhibitor ex vivo.
So sorry for explaining unclear in last version. We totally agree with you that LDH release from the cultured cells in vitro and from myocardium ex vivo are mediated through the different mechanism. LDH is released from ischemic hearts by disruption of cells (necrosis), whereas the isolated cells are NOT necrotic alone. The cellular membrane was broken in the late stage of apoptotic cardiomyocytes, evidenced by PI staining positive. Calcium overload induced by hypoxia/reoxygenation could cause the cellular necrosis as well in cultured cardiomyocytes. Despite of their different mechanism, level of LDH were here used as a readout of cellular membrane damage in vitro and ex vivo.
Q2:In the Langendorff experiment the authors may either used a constant-flow model and can easily calculate coronary resistance by perfusion pressure that must be recorded in the system to avoid edema, or used a pressure-constant model and can then easily calculate coronary resistant by measuring the flow. In any kind, if they use a pressure constant model an coronary resistance increases the have a stronger mechanic stress to the endothelium that may affect the readout. If the used a constant flow then an increase in coronary resistance means that the different readout may be affected by different oxygen supply. As the drug acts on Akt-Pathway that is also in endothelial cells active this is a critical point that needs to be addressed. The authors cannot simply say that they did not measure this. This is an important part of the study protocol.
A2:Yes, you are right. Your points are stimulating our version, creating a novel idea, and polishing the discussion. Indeed, the experiment conditions ex vivo are more complicated,including the effects of tested drug on the cardiac mycoytes, vascular endothelium, and vascular smooth muscle, etc. All of those will affect the drug-evoked cardiac contractile function and protection. As you pointed out, it is reasonable to suspect that AKT signaling in vascular endothelium could be a target of BAT07 in its cardiac protection since the present study had found the AKT signaling were suppressed by hypoxia/reoxygenation and BTA07 modulated the AKT signaling to display its cardiac protection in cultured cardiomyocytes. Those points are discussed in first paragraph in page 12. Thank you again.
By the way, we employed the model of constant pressure perfusion with a constant pressure of 80mmHg, and balloon pressure maintained at 0-10mmHg in Langendroff experiment. There was no cardiac edema obviously observed and the heart rates were maintained at above 300 times/min, suggesting that the perfused hearts were working very well within 210 min of the experimental procedures.
